# Contextual Fusion for Adversarial Robustness

## Abstract

Mammalian brains handle complex reasoning tasks in a gestalt manner by integrating information from regions of the brain that are specialised to individual sensory modalities. This allows for improved robustness and better generalisation ability. In contrast, deep neural networks are usually designed to process one particular information stream and susceptible to various types of adversarial perturbations. While many methods exist for detecting and defending against adversarial attacks, they do not generalise across a range of attacks and negatively affect performance on clean, unperturbed data. We developed a fusion model using a combination of background and foreground features extracted in parallel from Places-CNN and Imagenet-CNN. We tested the benefits of the fusion approach on preserving adversarial robustness for human perceivable (e.g., Gaussian blur) and network perceivable (e.g., gradient-based) attacks for CIFAR-10 and MS COCO data sets. For gradient based attacks, our results show that fusion allows for significant improvements in classification without decreasing performance on unperturbed data and without need to perform adversarial retraining. Our fused model revealed improvements for Gaussian blur type perturbations as well. The increase in performance from fusion approach depended on the variability of the image contexts; larger increases were seen for classes of images with larger differences in their contexts. We also demonstrate the effect of regularization to bias the classifier decision in the presence of a known adversary. We propose that this biologically inspired approach to integrate information across multiple modalities provides a new way to improve adversarial robustness that can be complementary to current state of the art approaches.

## 1 Introduction

### 1.1 Biological background

Current deep learning networks are designed to optimally solve specific learning tasks for a particular category of inputs (e.g., convolutional neural networks (CNNs) for visual pattern recognition), but are limited in their ability to solve tasks that require combining different feature categories (e.g., visual, semantic, auditory) into one coherent representation. Some of the challenges include finding the right alignment of unimodal representations, fusion strategy, and complexity measures for determining the efficacy of fused representationsBaltrusaitis et al. (2017). In comparison, biological systems are excellent in their ability to form unique and coherent object representations, which is usually done in the associative cortex, by linking together different object features available from different specialized cortical networks, e.g., primary visual or auditory cortices Gisiger et al. (2000); Pandya & Seltzer (1982); Mars et al. (2017); Rosen et al. (2017). This natural strategy has many advantages including better discrimination performance, stability against adversarial attacks and better scalability Gilad & Helmchen (2020). Indeed, if a classification decision is made based on a combination of features from different sensory categories, a noise or lack of information in one category can be compensated by another to make a correct decision. Furthermore, different types of sensory information can complement each other by being available at different times within a processing window. A good example may be human driving skill which relies on a combination of visual and auditory processing that helps to avoid mistakes and greatly enhances performance over only vision-based driving. Another example is insect navigation that depends both on visual and olfactory

information to minimize classification error and to identify objects more reliably across range of distancesStrube-Bloss & Rössler (2018).

Although it seems obvious that humans and animals base their classification decisions on the complex mixture of features from different modalities using specialized classifiers in each of them, this ability is still lacking in current state of the art machine learning (ML) algorithms. Problems include difficulty of training because of the lack of data sets combining different types of information, and suboptimal performance of generic deep learning networks vs specialized ones. Indeed, e.g., high performance of the CNNs designed for visual processing depends on their architecture that makes explicit assumptions that inputs are images, and the same network performs poorly for other types of data. Thus, there is a need to develop approaches that would combine strength of specialized networks with ability to integrate information across multiple streams as human and animal brain can do efficiently.

## 1.2 MULTI-MODAL FUSION

Multimodal fusion has been previously explored for hard classification problems. Proposed methods in literature include learning joint representations from unimodal representations that are derived using VLADGong et al. (2014), Fisher Vector representationsDixit et al. (2015), and deep features locally extracted from CNN's with various configurationsWu et al. (2015)Yoo et al. (2015)Shen et al. (2019). These approaches have been successfully applied for action, scene and event recognition, and object detection tasks.

In Zhou et al. (2014), the authors used features extracted from Alexnet pretrained on Places365 and Imagenet to show how internal representations of these networks perform for various scene and object centric datasets. Performance was not significantly superior to using a unimodal approach, however, some of the advantages were found to be related to reducing data bias. In Herranz et al. (2016) the authors demonstrated this by using combinations of object and scene features aggregated at different scales to build a more efficient joint representation that helped to mitigate dataset bias induced by scale. Our fusion approach aligns most closely with the methods proposed in these papers.

## 1.3 ADVERSARIAL ATTACKS

Image processing using deep convolutional neural nets (CNNs) has made historical leaps in the last decade Krizhevsky et al. (2012); He et al. (2016); Szegedy et al. (2015). However, the same convolutional networks are susceptible to small perturbations in data, even imperceptible to humans, that can result in misclassification. There have been two main approaches for investigating ANN robustness: adversarial machine learning and training data manipulation Ford et al. (2019). Although it has been proposed that adversarial and manipulation robustness can be increased through various mechanisms during the training phase, recent research has shown that these methods are mostly ineffective or their effectiveness is inconclusive Uesato et al. (2018) Geirhos et al. (2018); Athalye et al. (2018).

Fast Gradient Sign Method (FGSM)Goodfellow et al. (2014) is a popular one-step attack that is easier to defend compared to the iterative variants like Basic Iterative Method(BIM)Kurakin et al. (2016) or Projected Gradient Descent (PGD). Adversarial training and its variants are defense methods commonly employed for dealing with adversarial attacks. In Tramèr et al. (2017) the authors found that adversarial training is more robust with adversarial examples generated from white box attacks (attacks designed against the specifics of the underlying CNN architecture and weights) but it remains vulnerable to black box transferred examples (examples generated in architecture agnostic manner). To combat this, an ensemble model was proposed that combines adversarial examples created from different source models and substitute pretrained networks. In general, adversarial training on one type of attack does not generalise to other attacks and can compromise classification accuracy on clean, unperturbed data. For example, in Kurakin et al. (2017) the authors demonstrated that adversarial retraining on one step attacks do not protect against iterative attacks like PGD. Consequently, adversarial training with multi step attack is regarded as the state of the art method used for improving adversarial robustness for white box and black box attacks and was initially proposed in Madry et al. (2017). Recently, Wong et al. (2020) showed that single-step adversarial training with an attack similar to FGSM successfully yields models robust to white-box attacks, if the stepsizes of the attack's random and gradient step are appropriately tuned. Several other methods are

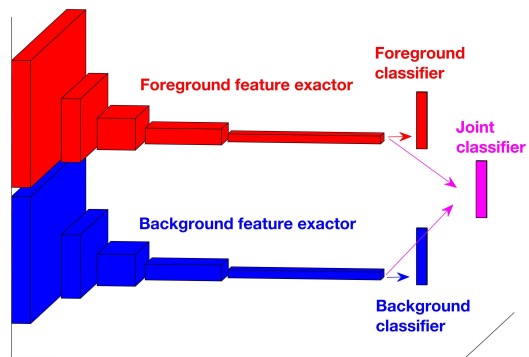

Figure 1: Cartoon of architecture: foreground, background, and joint classifiers.

proposed such as adversarial example detection, reconstructing adversarial inputs, network distillation etc. and are discussed in further detail in Yuan et al. (2018).

### 1.4 SUMMARY OF OUR APPROACH

In this paper, we describe the fusion of two data streams, one focused on background (context) and another focused on the foreground (object) image information, and we use different types of adversarial perturbations to evaluate the efficacy of the fused representation.

We explore the following main concepts:

- Adversarial attacks can have divergent effects on context feature space and object feature space.
- Utilizing combination of multiple modalities for the information processing can be an efficient method for combating adversarial attacks.
- Context features provide additional information to object-oriented data, and can be used to improve classification, especially during adversarial attacks.

## 2 METHODOLOGY

### 2.1 CONTEXTUAL FUSION

We developed three different image classifiers designed to extract foreground features, background features, or a fused version of both. Below we refer to these as the foreground, background, and joint classifiers, respectively. The distinction between the various classifiers is based on the underlying training data. The foreground classifier was trained on the object-centric Imagenet database, whereas the background classifier was trained on the scene-centric Places365 database similar to Zhou et al. (2014). Each classifier was built on the Resnet18 architecture He et al. (2016) with a final fully connected layer specific to the dataset (Figure 1). Only the 512 dimensional fully connected layer was finetuned for MS COCO Lin et al. (2015) or CIFAR-10 Krizhevsky & Hinton (2009). For the joint classifier, we adopted a late fusion strategy and concatenated features obtained from the foreground and background pre-trained networks. This 1024 dimensional representation was finetuned for the specific dataset.

### 2.2 DATASETS

MS COCO dataset is a large dataset with 1.6 million images that have multiple objects and multiple instances with overlapping contexts. We pruned the MS COCO dataset for images with fewer than two instances (<=2) of a single object so as to minimise cases of co-occurring context. We used available bounding boxes to constrain the percentage of foreground to be less than or equal to fifty percent of the total image, where foreground was defined by the area within the bounding box. This significantly reduced the dataset size but helped create the conditions for learning a representation

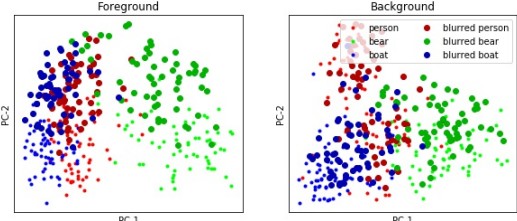

Figure 2: Effect of object blurring on background and foreground features. PCA projection to 2D-space is shown. Small and bright colored dots represent raw images while large and dark colored dots represent blurred images. Left, entire subspace of the foreground features moved (up in this example) in presence of blur. Right, after application of blur background features remain within the statistical subspace created by raw images. Each color represents a single image class. A filter with Gauss kernel and $\sigma = 5$ was used to blur images.

that had enough information for the background classifier to utilize. Our final dataset comprised of 24 classes with 7500 images. We used 75% of the dataset for training and the rest for testing. We also used CIFAR-10, which is a standard object recognition dataset with 10 classes. For experiments with CIFAR-10, we used the entire dataset with 50000 images for training and tested with the remaining 10000 images. All images from have been resized and cropped to 224x224 and normalized with CIFAR-10 mean and standard deviation.

## 2.3 ADVERSARIAL ATTACKS

We tested the classifiers with two different types of adversarial attacks to evaluate the benefits of fusion: Gaussian blur and FGSM. Gaussian blur was applied to the test set by convolving a portion of the image within the object bounding box given in the dataset with Gaussian kernel. Differing degrees of blur were created by varying the standard deviation of the kernel. Adversarial example attacks were generated using FGSM similar to Szegedy et al. (2013); Goodfellow et al. (2014). Here, small amounts of noise were added to the test images based on the gradient of the loss function with respect to the input. Following Goodfellow et al. (2014) we used the equation

$$\eta = \epsilon \cdot sign\left(\nabla_x J(\theta, x, y)\right)$$

where $\epsilon$ is a small real number, $J$ is the loss as a function of the parameters ($\theta$), the input ($x$), and the label $y$.

## 3 RESULTS

### 3.1 BLUR AFFECTS FOREGROUND AND BACKGROUND CHANNELS DIFFERENTLY

The impact of adversarial attacks can vary with the CNN architecture and the underlying training data Dodge & Karam (2016). Here, we used two distinct CNNs with the same Resnet18 architecture to serve as the background and foreground feature extractors. The background feature extractor was trained on the Places365 database Zhou et al. (2014), a scene-centric collection of images with 365 scenes as categories (e.g. abbey, bedroom, and library). The foreground feature extractor was trained on the object-centric Imagenet database with object categories (e.g. goldfish, English setter, toaster). The two different feature extractors represent the same images in different ways and have a different sensitivity to the same adversarial attack. We use pretrained models readily available in Pytorch.

Blur is a natural artifact common to many real-world acquired images. Therefore, it is critical for an image classifier to maintain robustness to blur. We created blurred images by convolution of the foreground of each image with a Gaussian kernel. This type of Gaussian blur is perceivable by humans, and for small standard deviations (e.g. $\sigma = 0.001$), the human visual system is able to perform a classification task with minimal mistakes. As we increase the amount of blur (e.g. $\sigma = 45$), both humans and neural networks tend towards chance level performance. In our approach, only the foreground pixels are modified leaving the context of the images intact.

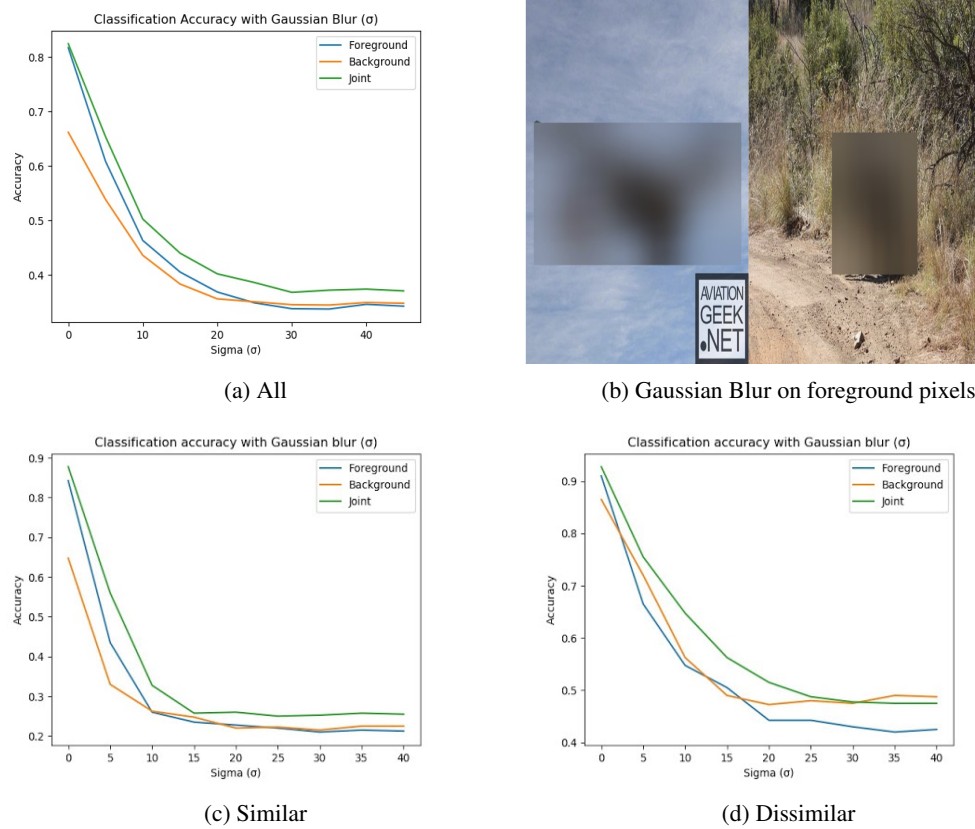

(a) All

(b) Gaussian Blur on foreground pixels

(c) Similar

(d) Dissimilar

Figure 3: Effect of Gaussian Blur on classification performance for MS COCO data. Panels a,c,d show classification performance for different levels of blur ($\sigma$). 'All' refers to the 24 classes from different supercategories that remain after downsizing the dataset. 'Dissimilar' and 'Similar' are eight classes randomly selected from the 24 classes such that the supercategories are either distinct or the same (see Methodology). b) Example of high Gaussian blur ($\sigma = 45$) on foreground pixels within bounding box.

We first tested effect of application of the Gaussian blur on images from the MS COCO dataset. The blur was applied to the bounding box area of the image, where main object was located. Images were processed by each of the two CNNs independently and features were extracted from the batch normalization layers. To illustrate effect of blur on the high-level foreground and background features, we visualized the feature space using PCA to reduce dimensionality for few representative image classes (see Figure 2). We found that blur had differing effects between the background (scene-centric) and foreground (object-centric) features. For the foreground features, blur caused a shift of the entire statistical representation subspace, i.e., all blurred images moved conjointly away from the non-blurred images (compare small vs large dots in Figure 2). However, for background features, blurred images were represented in the same statistical subspace as clean images. This shows, as one can expect, that blurring foreground (object) alone will have larger impact on the features extracted by the network focused on the foreground than on the features coming from the network that was train to recognize a background. This finding supports an idea that combined use of the foreground and background classifiers to process images in a multi-modal fashion may help to defend against adversarial attacks designed against specific image components. Specifically, if adversarial attacks affected only one channel of information (i.e. the foreground channel), then multi-modal integration could overcome these attacks.

### 3.2 Gaussian blur and the effect of contextual fusion

To directly test the hypothesis about increased robustness of the fusion based classifier against blur, we compared classification performance of the foreground, background, and fused classifiers on the images where foreground object was blurred. We found an increase in classification for all levels of $\sigma$ with the largest increase over foreground classifier for $\sigma > 5$ (Figure 3a).

Because, our joint fusion method depends on associating context with object, we also tested separately images with Similar and Dissimilar background context. To do this, we used the "supercategories" given in the MS COCO dataset to refine test sets into a "Similar test set" where all images came from different categories but the same supercategory (e.g. "dog", "horse", and "sheep" - all animals) and the "Dissimilar test set" where images come from distinct supercategories (e.g. "dog"-animal, "airplane"-vehicles, and "toilet"-indoor). This changes our problem to either a coarse grained classification e.g. classes from different supercategories like "Animals", "Vehicles", "Indoor", "Outdoor", or a much harder fine grained classification, e.g. classes from the same supercategory of "Animals".

An increase in classification was evident for each subset of the MS COCO dataset (Similar or Dissimilar) and almost all values of $\sigma$ (Figure 3c and Figure 3d). The largest improvements for the joint classifier were seen for moderate values of $\sigma$ and while using the Dissimilar test set (at $\sigma = 10$ the joint classifier outperforms the foreground classifier by 10%, Figure 3d). Thus, if all images in the test set contained different contexts (e.g. some were indoor images and others were outdoor images), then the joint classier performed much better than the individual ones. This finding was expected because contexts in the dissimilar test set contain more information specific to the underlying class. However, even when using the Similar test set, we found significant increases in classification over foreground alone. For example, classification with the joint classifier was 5% higher at $\sigma = 10$.

### 3.3 Gradient-based attacks

Adversarial attacks were constructed based on adding the right type of noise that maximizes the increase in the loss function, gradient-based attacks Szegedy et al. (2013). The magnitude of the added noise was quantified by $\epsilon$ (see Methodology and Goodfellow et al. (2014)). Here, we explored how well a fusion strategy can defend against them. Adversarial images were created from the full original images using FGSM method Goodfellow et al. (2014) applied to the foreground network, and performance was tested with the foreground, background and joint networks.

For MS COCO dataset with all 24 classes (the 'All' test set in Figure 3), we found that classification accuracy for the foreground classifier quickly dropped to chance with increasing $\epsilon$ - strength of the attack (Figure 4a). The background classifier degraded at a much slower rate compared to the foreground classifier. A joint classifier revealed somewhat intermediate level of performance across a range of the attack strengths. Importantly, joint classifier revealed the same level of performance as the foreground one for intact images, suggesting that our approach can overcome a common problem of the adversarial defense - degraded performance for intact images (Figure 4a). To reveal relative contribution of two networks (object-centric and scene-centric) to the joint classifier, we examined the weights of the trained joint classifier (Figure 4b) and found that the foreground and background networks influenced the joint decision almost equally. Qualitatively, this suggests that since the MS COCO dataset has object information embedded with semantically meaningful context, the background classifier was able to weigh in on the joint decision, thereby retaining performance above foreground network when it was affected by adversarial attack.

We next tested our fusion strategy on CIFAR-10 (Figure 4c). This dataset is much larger than the subset of MS COCO dataset we used, and it has established performance baselines in literature for different types of adversarial defensesWong et al. (2020), Yan et al. (2018). For the foreground classifier, we trained all layers of a Resnet18 on CIFAR-10, with weights initialized from a Resnet 18 pretrained on Imagenet. For the background classifier, we train only the last fully connected layer of Resnet18 with weights initialized from a Resnet18 pretrained on the Places365 dataset. This was done to ensure the background feature extractor represents scene-centric information, without overwriting to the object-based features in CIFAR-10. The joint classifier was a concatenated model of the foreground and background classifiers as described above such that only the last fully connected layer was trained and the model parameters of the foreground and background classifiers were retained. We found that the adversarial examples crafted using the foreground classifier also affected the joint classifier with its performance being only marginally above the performance of the foreground

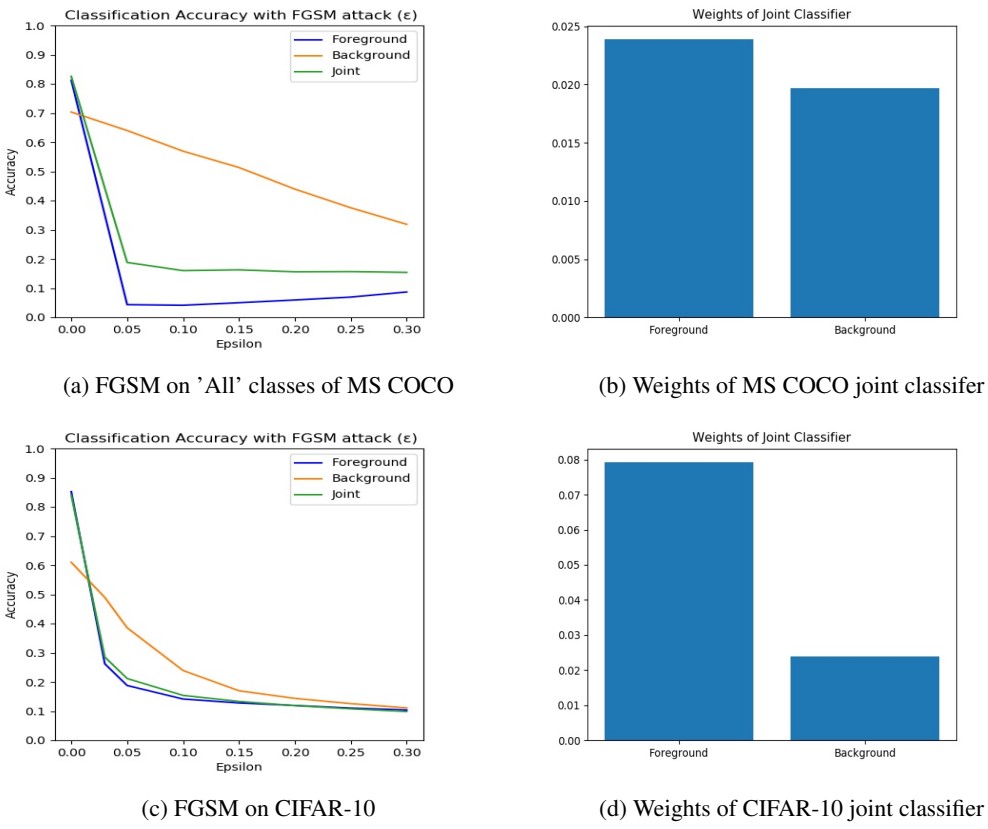

(a) FGSM on 'All' classes of MS COCO

(b) Weights of MS COCO joint classifer

(c) FGSM on CIFAR-10

(d) Weights of CIFAR-10 joint classifier

Figure 4: Effect of FGSM on MS COCO and CIFAR-10. a,c) FGSM attack on foreground classifier of 'All' categories from MS COCO (a) and CIFAR-10 (c). $\epsilon$ indicates strength of attack. b,d) Average absolute value of the weights from the last layers of the foreground and background networks to the joint classifier for MS COCO (b) and CIFAR-10 (d).

classifier. We further examined the weights of the joint classifier after training (Figure 4d) which revealed that the foreground features dominated the joint decision. This was most likely due to the fact that the background classifier significantly under-performed the foreground classifier at $\epsilon = 0$ ( 60% vs 85%). The last finding was not surprising because of the object-centric nature of the CIFAR-10 dataset, and the lack of meaningful background information. Thus, without significant input from background features, multimodal fusion failed to increase performance in FGSM attacks.

These results suggest that although adversarial examples do transfer across classifiers, a principled way to combine contextual information can be useful for adversarial robustness without the added cost of adversarial retraining and compromising accuracy on unperturbed data. This usefulness depended on two factors: a) the modalities needed to be sufficiently distinct; and b) the fusion mechanism needed to balance contribution of the information streams from different modalities. Below, we explored how this balance can be struck in an optimal way.

### 3.4    Regularization on known adversary

When it is known that the foreground is targeted by the attack, considerable gains can be made using the multi-modal method, even when compared to conventional methods like adversarial retraining. Inspired by the finding that improvement in the joint network is related to the relative weighting of background information, we employed regularization of the targeted foreground network to enhance the performance of the joint network. The equation below regularized the foreground weights of the joint network with L2 penalty determined by a tuneable hyperparameter $\alpha$.

$$L(a_i, t_i) = -\sum_{i}^{N} t_i log(\frac{e^{a_i}}{\sum_{j}^{C} a_j}) + \alpha|\theta_{fg}|^2$$

The first term was the standard cross entropy loss between the activation vector $a_i$ and the target vector $t_i$ for all N samples from C classes. The second term was the L2 penalty on the foreground weights, $\theta_{fg}$. This allowed the network to bias the classifier decision towards the background or the foreground based on the value of $\alpha$.

We also compared the performance of the regularized joint network with a standard method to combat FGSM attacks, adversarial retraining Madry et al. (2017). We found that tuning $\alpha$ had a strong effect on the performance of the joint network (Figure 5). The highest levels of regularization ($\alpha = 10$) enabled the joint classifier to perform close to the levels of the background classifier while retaining high performance on the clean test set. Since there was a known adversary on the foreground, the biasing mechanism described in the loss function equation above, can be used to weight the contextual decision of the background classifier more than the foreground. Surprisingly, the joint network with higher regulation outperformed the adversarial retraining strategy at all levels of $\epsilon$. It suggests that foreground regularization allows the user to select the degree of a more informative object-centric channel as opposed to a scene-centric channel that is less sensitive to the attack.

## 4    Conclusion

We present a novel method using semantic data fusion to increase robustness to adversarial attacks. By using features from distinct information streams, object-centered foreground and scene-centered background, we were able to maintain higher classification performance in the face of targeted adversarial attacks. We found that the degree of success for this method partially depends on the amount of variability in the background data. Thus, our method was more successful at maintaining robustness when objects came from distinct super categories of data with different and distinct backgrounds. Regularization of the foreground network enhanced this performance far above standard adversarial training strategies. In this work, we used a simple method of fusion - a single fully connected layer integrating the outputs of the object-centric and scene-centric networks. Our approach can be easily scaled by (a) adding other richer contexts and modalities (e.g., auditory, text) and (b) by implementing more sophisticated fusion layers inspired by neuro-scientific computational principles (e.g. recurrent networks, etc).

Our work may lead to better understanding of how knowledge is extracted from experience in biological networks and how brain dynamics are shaped by the development of a rich internal model

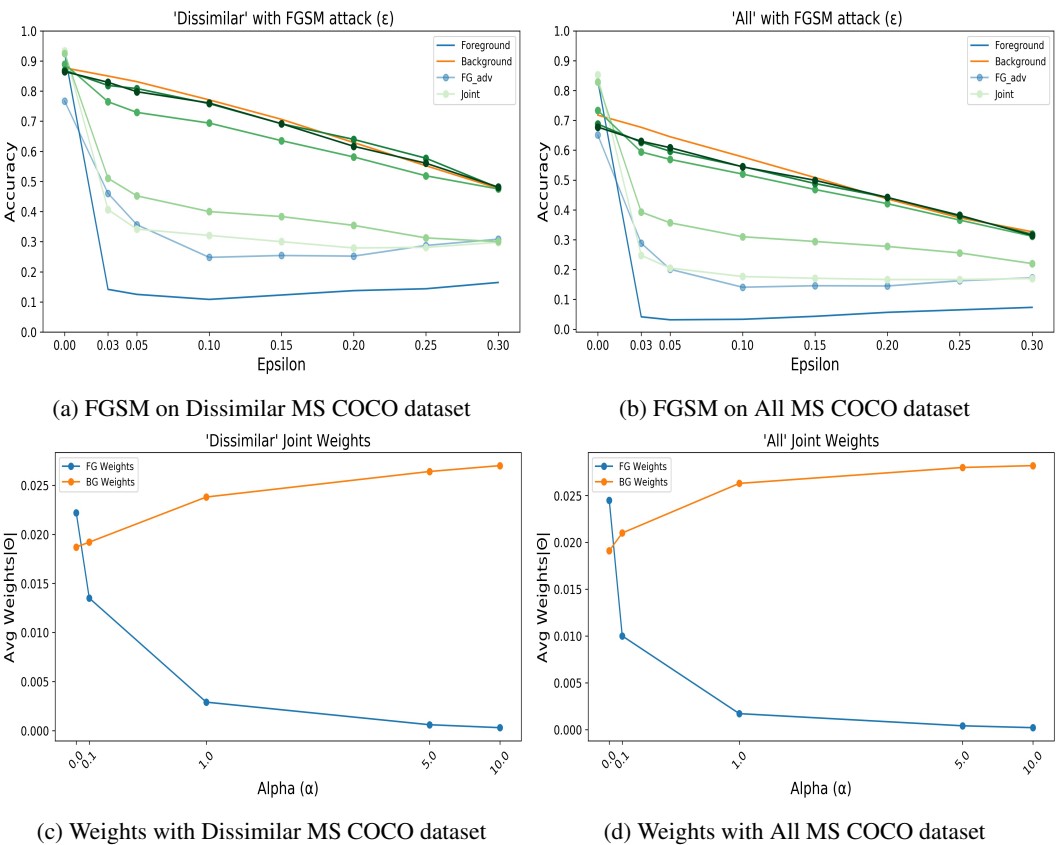

(a) FGSM on Dissimilar MS COCO dataset

(b) FGSM on All MS COCO dataset

(c) Weights with Dissimilar MS COCO dataset

(d) Weights with All MS COCO dataset

Figure 5: Regularization on the foreground weights of joint classifier. a)-b) Varying values of $\alpha$ on MS COCO Dissimilar and All categories are shown for the joint classifier in green. The range of $\alpha$ was between 0.1 and 10. Darker green colors indicate higher levels of $\alpha$. Examples for adversarial retraining were generated with an attack strength $\epsilon$ =0.3. c)-d) Average absolute value of foreground and background weights as a function of $\alpha$

of the world, including the ability to predict the outcomes of current situations and one's own actions in that context. The sophistication and complexity of the brain processing to combine different types of information streams to create internal model of the world is arguably the basis for "cognitive reserve", which is a significant factor in protection from age and disease related dementia, and so a deep understanding of how it is created and expressed is of high societal impact. Equally important, our study may provide insights into the algorithms that evolution has devised to make predictions about optimal behaviour based on the multimodal rich input from the world. This is a main issue in machine learning and using insight from biology will undoubtedly lead to advances in machine learning algorithms.

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
