# OpenReview forum: "Contextual Fusion For Adversarial Robustness"
_ICLR.cc/2022/Conference — ICLR 2022 Submitted_

### Official Review · Reviewer_FAzC · 2021-11-01

**Correctness:** 3
**Technical Novelty And Significance:** 1
**Empirical Novelty And Significance:** 2
**Recommendation:** 3
**Confidence:** 4

**Main Review:**

 I have the following concerns:

1. The idea for enhancing the adversarial robustness via foreground and background is not novel and has been studied in [A]. [A has similar conclusions but with a more challenging attack, e.g. PGD, instead of the FGSM.

2.  It is not clear why choosing MS-COCO dataset as a subject dataset. Commonly used datasets for adversarial attacks are the imagenet and CIFAR datasets. Why not use imagenet dataset?

3. Why choose Gassuain blur as a perturbation? Note that, recent work [B] has studied the adversarial attack from the angle of motion blur. In contrast to Gaussian blur, the adversarial motion blur could fool DNNs via gradient information like the traditional noise attack.

4. All figures show obvious distortions and there are a lot of typos. This work may be a rush to the deadline.

[A] Towards Robustness against Unsuspicious Adversarial Examples.
[B] Watch out! Motion is Blurring the Vision of Your Deep Neural Networks. NeurIPS 2020.

**Summary Of The Paper:**

This work proposed to enhance the robustness of DNNs by fusing context information from the background. It first studied the blur effects to the foreground and background-based DNNs and observed that fusing the two information helps accuracy improvements under different blur effects. Then, it further extends to the adversarial attacks via FGSM, and observes the advantages of using background information on MSCOCO and CIFAR-10 datasets. Finally, it proposed a regularization method to reweigh the foreground-related weights during training.

**Summary Of The Review:**

Overall, the main concerns of this work are the novelty and unclear experimental setups.

---

### Official Review · Reviewer_GiW7 · 2021-11-02

**Correctness:** 2
**Technical Novelty And Significance:** 1
**Empirical Novelty And Significance:** 1
**Recommendation:** 3
**Confidence:** 4

**Main Review:**

There are three major weaknesses in this paper:

1. The pre-trained models selected for recognizing foreground and background are not convincing. There is no proof that the one trained on ImageNet can be used as a foreground objects detector. If you ever checked the detailed class labels of ImageNet, you will know there are many classes that are similar to Place-365, vice-versa.

2. The novelty is limited. The method can be seen as an ensemble of different models. Moreover, the FGSM is only targeting the foreground module, leaving the background module untouched.

3. The experiments are weak, including the selection of datasets and attack methods.

**Summary Of The Paper:**

The paper tackles the adversarial example problem. The authors propose an approach that is motivated by the way biological systems employ multi-modal information to recognize category of objects. Specifically, the approach combines two pre-train models, that are excepted to focus on foreground and background, respectively. Then the foreground module is fine-tuned for downstream tasks while the background module is left unchanged. The authors demonstrate that they obtain better performance against blur and FGSM.

**Summary Of The Review:**

Due to the weaknesses mentioned, I recommend to reject this paper.

---

### Official Review · Reviewer_mH95 · 2021-11-02

**Correctness:** 2
**Technical Novelty And Significance:** 1
**Empirical Novelty And Significance:** 1
**Recommendation:** 3
**Confidence:** 4

**Main Review:**

Strengths:
* Robustness to adversarial examples is a hot topic within the ML community. However, relatively less attention has been spent on the explorations of fusion based models against adversarial attacks. Therefore, I believe that the main focus of this paper is very relevant to the ICLR community.

Weaknesses / discussion questions:
* The paper exceeds the page limitation, which is not fair for other submitted manuscripts within the page limitation.
* The integration of fusion and adversarial learning should be a very interesting topic to be studied with. The contribution of this paper seems to be making explorations within this domain. Then the authors should point it out explicitly, instead of saying “Summary of our approach” in section 1.4. And, the paper is not making clear what contributions are novel, and what is from existing work. I think it would be better to separate a Related Work section from the Introduction, and describe more prior work of making fusion networks against adversarial robustness (e.g., [1]) and the differences between your method and other methods for clarity. Then, in the Methodology section, authors describe how to leverage and fuse pre-trained models, and test its performance against adversarial attacks. The authors are suggested to highlight your **proposed** method, otherwise, it is more like a technical report with lack of novelty. In my opinion, the contribution of this work is not enough.
* Authors are suggested to gather more prior work, re-design the experimental settings, compare with other related methods, and demonstrate its performance against stronger attacks (e.g., PGD, CW and AA).
* The paper seems to be written in a rush. There are several format errors, typos, grammatical errors, and sentences that fail to convey ideas. The writing of the paper needs polishing. All the citations are mixed with main text, making the paper not easy to follow. Figures are blurred, for examples, the text in figure 4 & 5 is not clear and being stretched.

[1] Yu et al, Towards Robust Training of Multi-Sensor Data Fusion Network Against Adversarial Examples in Semantic Segmentation, IEEE ICASSP, 2021.

**Summary Of The Paper:**

In this paper, authors studies the problem of adversarial training and tries to leverage a fusion-based method against adversarial attacks. This method fuses features from foreground and background extracted by pre-trained models and test its performance against both Gaussian blur and gradient-based attacks. The authors claim three main explorations:
* Exploring the effects of adversarial attacks on both context and object feature space.
* Exploring the benefits of fusing different modalities against adversarial attacks.
* Exploring the benefits of context features.

**Summary Of The Review:**

This paper studies the problem of adversarial training and tries to leverage a fusion-based method against adversarial attacks. The integration of fusion and adversarial learning should be a very interesting topic to be studied with. However, I have concerns about the technical quality, the novelty of the manuscript, and the violation in page limitation. All of these lead me to recommend its rejection.

---

### Official Review · Reviewer_yT4G · 2021-11-02

**Correctness:** 2
**Technical Novelty And Significance:** 1
**Empirical Novelty And Significance:** 1
**Recommendation:** 1
**Confidence:** 5

**Main Review:**

The Authors of the paper have not properly quantifiedWhile the intention of the paper is good, this paper unfortunately does not meet the bar/standard for an ICLR submission, and may also be reporting mis-leading results given the correctness of how the attacks were computed.

This papers main weaknesses are:
* No use of errorbars or confident intervals in the adversarial attacks or blur based perturbation
* Section 2.3: Gaussian blur is not a type of Adversarial Attack; it is an out-of-distribution type of image distortion/manipulation.
* Authors should have used PGD based attacks to strengthen their claims.
* Authors should expand on using different CNN-based architectures.
* Clarity: it is not obvious how the fore/background networks are partitioned into separate streams and then unified to be fully end-to-end differentiable.
* Most importantly: I am not convinced the results here are veridical given the way the adversarial attacks have been made. It seems by figure 1 that the fusion network is not end-to-end differentiable. If it is not end to end differentiable, then how is the gradient computed for the FGSM attack to actually maximize the loss? (Maybe I missed something?)

Overall the idea of using parallel foreground/background networks is appealing for adversarial robustness, but there are still some missing works I encourage the authors to look into:

* Putting visual object recognition in context. Zhang, Tseng & Kreiman. CVPR 2020.
* Human peripheral blur is optimal for object recognition. Pramod, Kitti & Arun. ArXiv 2020.
* Emergent properties of foveated perceptual systems. Deza & Konkle, ArXiv 2021.

The figures in general could all use more work.


**Summary Of The Paper:**

This paper presents a very preliminary first step into designing a foreground/background CNN that is robust to adversarial attacks.

**Summary Of The Review:**

While I find the idea interesting, and I like the direction the authors are going -- this work is still quite preliminary and needs more work.

---

### Decision · Program_Chairs · 2022-01-20

**Decision:**

Reject

**Comment:**

This manuscript proposes an information fusion approach to improve adversarial robustness. Reviewers agree that the problem studied is timely and the approach is interesting. However, note concerns about the novelty compared to closely related work, the quality of the presentation, the strength of the evaluated attacks compared to the state of the art, among other concerns. There is no rebuttal.